# Protecting Apricot Orchards with Rain Shelters Reduces Twig Blight Damage Caused by *Monilinia* spp. and Makes It Possible to Reduce Fungicide Use

Laurent Brun [1,*], Freddy Combe [1], Christophe Gros [1], Pascal Walser [2] and Marc Saudreau [2]

[1] INRAE, UERI, 460 Chemin de Gotheron, 26320 Saint-Marcel-lès-Valence, France; christophe.gros@inrae.fr (C.G.)

[2] INRAE, Université Clermont Auvergne, UMR 547 PIAF, 63178 Aubière, France; pascal.walser@inrae.fr (P.W.); marc.saudreau@inrae.fr (M.S.)

* Correspondence: laurent.brun@inrae.fr; Tel.: +33-4-32-72-21-02

**Abstract:** Blossom and twig blight, caused by *Monilinia* spp., is the main disease in apricot trees. In this study, we installed transparent rain shelters in apricot orchards to study their influence on the modification of the microclimate at the level of the tree canopy and on the reduction in moniliosis damage in twigs. Rain shelters significantly reduced the leaf wetness time measured within the foliage compared to the unsheltered trees (a reduction of between 43% and 67%). However, very few differences were observed in the daily averaged air temperature (up to 6%) and daily averaged air relative humidity (up to 1%). In the first experiment, on the apricot variety Bergarouge® (CEP Innovation, Lyon, France), moniliosis damage on twigs in the absence of phytosanitary protection was reduced by up to 62% for the trees provided with rain protection compared to the trees that did not receive rain shelters. A second experiment, involving five apricot tree varieties, made it possible to verify that fungicide protection could be reduced for the trees protected by rain covers, reducing moniliosis damage on twigs compared to full fungicide protection combined without rain protection. Finally, a third experiment comprising two apricot tree varieties has shown that in organic orchards, rain protection provides protection against moniliosis (twig blight) that is equivalent to an organic farming fungicide protection programme based on the use of copper sulphate and calcium polysulphide.

**Keywords:** blossom blight; twig blight; *Monilinia* spp.; apricot; rain shelters; leaf wetness duration; fungicide use reduction

## 1. Introduction

Among the many pathogens capable of attacking *Prunus* trees, brown rot caused by different species of *Monilinia* is one of the most important economic factors limiting the production of stone fruit around the world [1,2]. In stone fruit, *Monilinia* spp. are able to infect various plant organs, causing blossom blight, twig blight and brown rot in immature and mature fruits. The two main species responsible for attacks on flowers and twigs in *Prunus* trees are *Monilinia laxa* (Aderhold and Ruhland) Honey and *Monilinia fructicola* (G. Winter). Apricot is the crop that is most susceptible to blossom and twig blight, followed in order by prune, sweet cherry, peach, sour cherry and plum trees [3]. *M. laxa* can cause infections in apricot blossom, twigs and fruit. The first two are of the greatest concern, especially in organic production, causing losses of up to 90% in southern France [4].

During winter, *M. laxa* and *M. fructicola* are preserved as mycelium in cankers on twigs infected the previous year and in mummified fruit which is hanging from branches or has fallen to the ground. For *M. fructicola*, there may also be the formation of apothecia in mummified fruit, which then produce ascospores. Conidia and ascospores constitute the primary inoculum and can be transported by wind, rain and insects [1]. When the

conditions are humid, the flowers can be infected, with the mycelium progressing via the peduncle to reach the twigs and cause a canker there, which can lead to the apical section of the twig drying out [2]. All parts of the flower can serve as the first infection site. There is evidence that fully open flowers are the most susceptible to infection [3]. Luo et al. [5] demonstrated that prune flowers have been shown to be the most sensitive to contamination by *M. fructicola* when wide open (BBCH Stage 65) [6]. In the orchard, the period of sensitivity of apricot flower buds to contamination by *M. laxa* begins at the 'sepals open' stage (BBCH stage 57), increases until the 'full flowering' stage (BBCH stage 65) and ends at the 'flowers fading' stage (BBCH stage 67) [6–8].

The presence of humid conditions at the time of flower susceptibility is crucial in paving the way for contamination by *Monilinia* spp. Tamm et al. [9] have shown in a growth chamber experiment the effects of petal wetness duration and temperature on the development of blossom blight caused by *M. laxa* in cherry trees (*Prunus avium* L.). They report that disease incidence increases with the duration of flower wetting. Luo et al. [5] have shown similar results for *M. fructicola* on plum flowers (*Prunus domestica* L.). In an experiment with potted apricot trees installed in the orchard over short periods of 24 to 48 h, Tresson et al. [7] confirmed that the duration of wetting was indeed an explanatory parameter for the incidence of *M. laxa* on flowers, but that the amount of rain (in mm) was a more explanatory parameter than the duration of wetting. Therefore, they proposed a model making it possible to estimate, for each rainy episode, the incidence of *M. laxa* on flowers according to rainfall and temperature for an orchard in which the aerial inoculum was between 50 and 200 conidia per $m^3$ of air [8]. Rain could also play a role in the dispersal of conidia through a splashing effect. In a study conducted in California, Corbin and Ogawa [10] showed that there was a greater dispersal of *M. laxa* conidia in the air during rainy periods than in dry periods.

In some perennial crops, rain shelter systems have shown great effectiveness in reducing the incidence of many fungal diseases whose development requires a certain period of wetting in trees. For example, in Norway, Borve and Stensvand [11] showed that the installation of transparent rain shelters on cherry trees during rainy periods from flowering to harvest made it possible to avoid fungicide protection while producing a relatively healthy harvest. In fact, covered trees presented on average only 3.4% of rotten cherries, mainly due to *Monilinia laxa* and *Botrytis cinerea* (Pers.). In contrast, unprotected trees received between three and six fungicide treatments from flowering to harvest and averaged 16.5% of rotten cherries [11]. In France, the rain shelters offered by Filpack® were evaluated from 2010 to 2015 on Braeburn and Gala apple varieties. In the absence of fungicide protection, the development of scab due to *Venturia inaequalis* (Cooke) G. Winter was greatly reduced in sheltered trees compared to non-sheltered trees. Even in years with high scab pressure, such as 2012 and 2013, the symptoms on the leaves and fruits were very limited, with less than 3% of scabbed leaves and less than 1% of scabbed fruits in sheltered trees compared to 94% to 100% of scabbed leaves and 63% to 100% of scabbed fruits in non-sheltered trees [12]. A study conducted on table grapes in southern China's Yunnan Province showed the strong potential of rain shelters in reducing the development of major vine diseases. For two years, under the rain shelter, the severity of downy mildew on grapes due to *Plasmopara viticola* (Berk. and Curtis) Berl. and de Toni was reduced by an average of 81% compared to the vines receiving fungicide protection, and by 94% compared to the control vines without fungicide protection. The severity of bunch rots due to *Colletotrichum gloeosporioides* and *C. petrakii* B. Sutton were reduced under rain shelters by 85% and 69%, respectively, compared to the vines receiving fungicide protection, and by 93% and 90% compared to the control vines without fungicide protection [13].

In this study, we wanted to evaluate the advantages of installing rain shelters in orchards to protect apricot trees from moniliosis outbreaks on flowers and twigs. The first objective of this study was to evaluate the influence of transparent rain shelters on the modification of the microclimate at the level of the tree canopy and on the reduction in moniliosis damage to the twigs. The second objective was to test the effectiveness of protection strategies combining rain shelters and reduced fungicide protection against moniliosis in both conventional and organic production.

## 2. Materials and Methods

### 2.1. Experimental Orchards

The three experimental orchards were located at the INRAE-UERI experimental station in Saint-Marcel-lès-Valence (Drôme) in France's Middle Rhone Valley. This is a continental area with Mediterranean summer influences.

All the apricot varieties used in the three orchards were grafted on Montclar® (4238) (Agri Obtentions, Guyancourt, France) rootstock. Trees were irrigated using localised micro-sprinkling under the tree foliage. This irrigation system avoids wetting the leaves and fruits during irrigation. There is no tree irrigation when the apricot trees bloom.

Transparent plastic rain shelters in high-density polyethylene (Anisolar, Filpack, Vitrolles, France), comprising two 1.5 m wide strips connected on a ridge cable above the tree row, were installed in the 'Rain shelter' mode of the three experimental trials, providing 3 m wide protection at a height of 3.5 m. The rain shelters were unrolled before the beginning of the sensitivity of flower buds to moniliosis contamination (Table 1). They were folded up in July after fruit harvesting.

**Table 1.** Period of sensitivity of apricot blossoms to contamination by *Monilinia* spp. (according to Tresson et al. [7] and Brun et al. [8]) and rainfall over this period for the three trials.

| Experimental Orchard | Year | Variety | Period of Sensitivity to Contamination | | Rainfall over Sensitive Period (Cumulative in mm) |
|---|---|---|---|---|---|
| | | | Start (BBCH 57) | End (BBCH 67) | |
| Rain shelter protection evaluation | 2014 | Bergarouge® | 3 March 2014 | 24 March 2014 | 25.5 |
| | 2015 | Bergarouge® | 18 March 2015 | 4 April 2015 | 28.5 |
| Rain shelter and fungicide use reduction | 2016 | 5 varieties | 4 March 2016 | 31 March 2016 | 58.5 |
| | 2017 | 5 varieties | 2 March 2017 | 21 March 2017 | 14.0 |
| | 2018 | 5 varieties | 12 March 2018 | 5 April 2018 | 67.0 |
| Organic orchard without copper | 2021 | Tom Cot® | 25 February 2021 | 22 March 2021 | 10.0 |
| | | Vertige | 25 February 2021 | 22 March 2021 | 10.0 |
| | 2022 | Tom Cot® | 25 February 2022 | 25 March 2022 | 22.0 |
| | | Vertige | 28 February 2022 | 28 March 2022 | 22.0 |

### 2.1.1. Experimental Orchard 'Rain Shelter Protection Evaluation'

This orchard, planted in 1998, involved Bergarouge® (2914) apricot trees, a cultivar with a high susceptibility to blossom and twig blight [4]. The distance between trees was 3.5 m, with a 5 m gap between rows. In October 2013, the trees, initially pruned in 'gobelet' style, were pruned to form a rectangular parallelepiped 3 m high and 2 m wide to ensure trees could be protected from rain under the shelters. Control trees were pruned in the same fashion. Treatments ('Rain shelter protection' and 'Control without protection') were replicated twice with a block design. The experimental system comprised 12 trees under rain shelters and 12 trees without rain shelters.

The rain shelters were unrolled on 24 February 2014 and 3 March 2015. No fungicides were applied to control blossom and twig blight in 2014 and 2015.

2.1.2. Experimental Orchard 'Rain Shelter and Fungicide Use Reduction'

This orchard was established in January 2015. It has five varieties, including Bergeron (660), Bergeval® (3950) (CEP Innovation, Lyon, France), Shamade (3902) and Anegat (4481), which are considered susceptible to moniliosis on flowers and twigs, as well as Frisson (2821), which is considered very susceptible to moniliosis [4]. Each elementary plot consists of a row of eight trees of the same variety, with a planting distance of 2.5 m between trees and 4 m between rows. The trees were trained in trellised 'palmettes' so as to form a fruit wall about 1 m wide. The 'protection system' study factor is made up of two modes, one without rain shelters with classic fungicide protection (called 'Fungicide reference'), and one with rain covers combined with reduced fungicide protection (called 'Rain shelter'). Treatments were replicated twice with a block design. The trial involved 80 trees under rain shelters and 80 trees without rain shelters.

The rain shelters were unrolled on 11 February 2016, 21 February 2017 and 5 March 2018. During the period when flower buds are sensitive to contamination, preventive fungicide protection was used, taking into account rainy episodes announced in weather forecasts. Fungicide treatments were renewed when new flowers opened after the previous application. The reduction in fungicide protection for the 'Rain shelter' mode consisted of removing the last fungicide application positioned at the end of the period of flower sensitivity to contamination by moniliosis (Table 2). No fungicide protection against moniliosis on flowers was conducted in 2015 because there were no flowers on the trees during the first year of growth in the orchard.

**Table 2.** Fungicide spraying schedules against twig blight in the modes 'Fungicide reference' and 'Rain shelter' in the 'Rain shelter and fungicide use reduction' trial in 2016, 2017 and 2018.

| Date | Active Ingredient and Application Rate (g/ha) | Product Name | Company |
|---|---|---|---|
| 2016 | | | |
| 2 March 2016 | Cyprodinil (75 g/ha); Fludioxonil (50 g/ha) | Switch | Syngenta France |
| 15 March 2016 | Iprodione (750 g/ha) | Rovral Aqua Flo | BASF Agro |
| 30 March 2016 * | Difeconazole (37.5 g/ha) | Difcor 250 EC | Globachem NV |
| 2017 | | | |
| 7 March 2017 | Cyprodinil (75 g/ha); Fludioxonil (50 g/ha) | Switch | Syngenta France |
| 17 March 2017 | Difeconazole (37.5 g/ha) | Difcor 250 EC | Globachem NV |
| 2018 | | | |
| 13 March 2018 | Cyprodinil (75 g/ha); Fludioxonil (50 g/ha) | Switch | Syngenta France |
| 24 March 2018 | Cyprodinil (75 g/ha); Fludioxonil (50 g/ha) | Switch | Syngenta France |
| 5 April 2018 * | Difeconazole (50 g/ha) | Difcor 250 EC | Globachem NV |

* Fungicide treatment conducted only in 'Fungicide reference' mode.

2.1.3. Experimental Orchard 'Organic Orchard without Copper'

This experimental system was established in January 2020 and is managed according to the French rules for organic production. It includes two varieties, Vertige (3845), which is considered susceptible to moniliosis on flowers and twigs, and Tom Cot® (2669) (Cot International, Bouillargue, France), which is considered fairly insensitive [4]. The elementary plots consist of five rows of six trees, i.e., thirty trees of the same variety. Planting distances between trees are 2.5 m, with 4 m between rows. The 'protection system' study factor comprised three modes as follows: (i) an 'Organic farming reference' mode in which the trees are grown 'gobelet' style and for which a classic organic farming fungicide protection programme against moniliosis is applied, (ii) a 'Control' mode in which the trees are grown 'gobelet' style without fungicide protection against moniliosis provided, and (iii) 'Rain shelter' mode in which the trees are trained in a 'palmette' 1 m wide under rain shelters

and for which no fungicide protection against moniliosis is provided. Both varieties were included in all three modes, with treatments replicated twice with a block design. The experiment involved 360 trees, including 120 trees under rain shelters.

Rain shelters were unrolled on 18 February 2021 and 23 February 2022. During the period when flower buds are sensitive to contamination, fungicide protection in the 'Organic farming reference' mode was conducted (i) either preventively with copper sulphate taking into account rainy episodes announced in weather forecasts, or (ii) at the end of the rainy period with calcium polysulphide (Table 3).

**Table 3.** Fungicide spraying schedules against twig blight in the 'Organic farming reference' mode of the 'Organic farming without copper' trial in 2021 and 2022.

| Date | Active Ingredient and Application Rate (g/ha) | Product Name | Company |
|---|---|---|---|
| 2021 | | | |
| 23 February 2021 | Copper sulphate (6250 g/ha) | Bordeaux mixture RSR Disperss | UPL Europe |
| 4 March 2021 | Copper sulphate (1000 g/ha) | Bordeaux mixture RSR Disperss | UPL Europe |
| 10 March 2021 | Copper sulphate (1000 g/ha) | Bordeaux mixture RSR Disperss | UPL Europe |
| 2022 | | | |
| 23 February 2022 | Copper sulphate (6250 g/ha) | Bordeaux mixture RSR Disperss | UPL Europe |
| 14 March 2022 | Calcium polysulphide (6080 g/ha) | Curatio | Andermatt France |
| 17 March 2022 | Copper sulphate (1000 g/ha) | Bordeaux mixture RSR Disperss | UPL Europe |

### 2.2. Twig Blight Assessment

For the 'Rain shelter protection evaluation' and 'Rain shelter and fungicide use reduction' trials, the damage assessment of *Monilinia* spp. on twigs was carried out by counting the number of twigs dried out by *Monilinia* spp. per tree.

For the 'Organic orchard without copper' trial, the twig blight assessment consisted of an observer visually evaluating *Monilinia* spp. infection in the tree. This evaluation was carried out 30 days after full blossom. For each tree, the observer visually estimated the ratio between the total length of flower-bearing twigs dried out by *Monilinia* spp. and the total length of flower-bearing twigs. This meant that the scale used was from 0 to 100% (total infection).

In all trials, once scoring was completed, all moniliated twigs were cut and removed from the orchard. This prophylactic action makes it possible to restore each of the different modes to a primary inoculum level of the same order of magnitude for the year n + 1.

### 2.3. Identification of the Monilinia Species Responsible for Moniliosis on Flowers and Twigs

The identification of *Monilinia* species began with the collection of orchard samples of flowers and moniliated twigs. Moniliated flowers were placed directly in a Petri dish containing a PDA medium (potato dextrose agar, Conda Laboratory, Spain; 9.75 g of PDA medium for 250 mL of deionised water). To take isolations from the twigs, a portion of the twig was taken from under the bark at the boundary between the healthy and necrotic zones and placed in a Petri dish. After 10 days of growth of the mycelium, identification was made according to the morphological characters of the mycelium defined by the Lane's synoptic key [14].

### 2.4. Climatic Conditions and Canopy Microclimate

Climatic data were recorded by an Enerco 516i meteorological station (Cimel Electronique, Paris, France) from INRAE's national Agroclim network, located on grassland between 300 and 900 m from the experimental orchards.

To assess the effect of rain shelters on microclimatic conditions during the 'Rain shelter protection evaluation' and 'Rain shelter and fungicide use reduction' trials, air temperature, air relative humidity and leaf wetness were recorded within tree canopies. Air temperature and relative humidity were measured using iButton® temperature/humidity loggers (THB—DS1923, Dallas Semiconductor, Dallas, TX, USA). All loggers were protected in well-ventilated white shelters of 950 cm$^3$ to prevent them from direct sunlight and provide airflow. Leaf wetness duration (LWD) was measured using home-made artificial leaves built from a rectangular electrical insulator comprising two interdigitated electrodes. The LWD estimate is based on determining the electrical resistance on the surface of the sensor [15]. The estimation of the LWD was conducted as follows: First and prior to the experiment, the resistivity (in ohms) of the sensor was estimated in controlled conditions for dry and wet conditions. A threshold value was then obtained that enables us to indicate whether liquid water is present (LWD is set to 1) or not (LWD is set to zero) at the sensor surface. In practice and in the experiment, the LWD was estimated by converting the electrical resistance measured by the sensors in 0 and 1 according to the threshold value. This value was multiplied by acquisition time (for instance, 10 min) to obtain the LWD. The procedure was applied to the entire dataset to obtain the LWD. The values were summed to obtain the monthly values.

For the 'Rain shelter protection evaluation' trial, three trees per block were equipped with a LWD sensor, and two trees per block were equipped with a THB sensor. Each LWD sensor was located at a height of 1.8 m and 0.6 m from the tree trunk following the row axis and on the north side of the tree. THB sensors were attached to the north side of the trunk at a height of 1.8 m. Data were collected from February to October every 10 min during 2014 and 2015. For the 'Rain shelter and fungicide use reduction' trial, six LWD and two THB sensors per mode were mounted within tree canopies. For each mode, sensors were located at a height of 1.8 m and distributed around two trees (three sensors per tree for LWD and one THB per tree) following the north-western, north-eastern and southern directions relative to the trunk. Data were collected from February to September every 10 min in each year.

### 2.5. Data Analysis

The number of twigs per tree dried out by *Monilinia* spp. and the proportion of necrotic twigs showing symptoms of *Monilinia* spp. were subjected to an analysis of variance (ANOVA) using Statgraphics Plus 5.1 software (Manugistics, Rockville, MD, USA). The level of significance was set at 5% for all the statistical tests. The normal distribution of ANOVA residuals was checked using the Shapiro–Wilk test [16], and the independence of ANOVA residuals and the intra-treatment variance equality (homoscedasticity hypothesis) were visually checked using the residuals/predicted values graph. When residual standard deviation increased with the increment of predicted values, data were log-transformed before ANOVA [17]. Mean comparisons were conducted using the Newman–Keuls test.

For the 'Rain shelter protection evaluation' trial ('Organic orchard without copper'), each year, the number of moniliated twigs per tree (the percentage of necrotic twigs) was subjected to a two-factor variance analysis (treatment factor and block factor).

For the 'Rain shelter and fungicide use reduction' trial, each year, the number of moniliated twigs per tree was subjected to a two-factor variance analysis (treatment factor and variety factor). In the absence of significant treatment x variety interaction, the results are presented for the average of the five varieties.

The entire leaf wetness duration dataset was separated into leaf wetness duration due to rain and dew by considering rain events and clear night sky conditions, respectively, identified from the meteorological data.

## 3. Results

### 3.1. 'Rain Shelter Protection Evaluation' Trial

In 2014, the period of sensitivity of apricot blossoms to contamination extended from 3 March to 24 March (Table 1). Over this period, the total wetness duration was significantly reduced (56%) by rain shelters compared to the unsheltered section of the orchard. The reduction was greater during dew events (60%) than during rain events (44%). The air temperature and air relative humidity were not affected by the rain shelters (Table 4). In 2015, the period of sensitivity to contamination was later in the season, extending from 18 March to 5 April (Table 1). Over this period, the total wetness duration was significantly reduced by 41% by rain shelters compared to the unsheltered section of the orchard. The reduction was more significant during rain events (53%) than during dew events (17%). The air temperature and air relative humidity were not affected by the rain shelters (Table 4).

**Table 4.** Averaged relative changes in leaf wetness duration, air temperature and air relative humidity in the tree canopy between sheltered trees and unsheltered trees during periods of susceptibility to infection (March) for the '*Rain shelter protection evaluation*' trial for 2014 and 2015. Changes for the variable X are estimated as $(X_{sheltered} - X_{unsheltered})/X_{unsheltered}$.

| Year | Air Relative Humidity | Air Temperature | Changes in Leaf Wetness Duration | | |
| --- | --- | --- | --- | --- | --- |
| | | | **Rain** | **Dew** | **Total** |
| 2014 | −1% | 1% | −44% | −60% | −56% |
| 2015 | −0.5% | 1% | −53% | −20% | −43% |

The identification of the *Monilinia* species responsible for the symptoms on flowers and twigs in the 'Rain shelter protection evaluation' orchard showed that *M. laxa* accounted for 100% of isolates in 2014, but only 55% in 2015. In this orchard, the 45% presence of *M. fructicola* in 2015 was unique, as no other values of a similar scale were observed across the site's apricot orchards from 2014 to 2022.

In the absence of fungicide protection, high levels of moniliosis damage were recorded in 2014 and 2015 in the 'Control without protection' mode, with several hundred dried twigs per tree observed (Table 5). In 2014, the 'Rain shelter protection' mode provided a slight reduction (26%) in the number of twigs dried out by *M. laxa*. In 2015, the reduction in the number of twigs dried out by *Monilinia* spp. was greater, reaching almost 62% (Table 5).

**Table 5.** Number of moniliated twigs (dried out by *Monilinia* spp.) per tree and percentage reduction in the number of moniliated twigs in 'Rain shelter protection' mode compared to 'Control without protection' mode in the 'Rain shelter protection evaluation' trial in 2014 and 2015.

| Date | Number of Moniliated Twigs per Tree | | Probability (*p*) | Reduction Percentage |
| --- | --- | --- | --- | --- |
| | **Control without Protection** | **Rain Shelter Protection** | | |
| 15 April 2014 | 368.5 | 274.0 | 0.0194 * | 25.6% |
| 29 April 2015 | 287.0 | 109.8 | 0.0347 * | 61.7% |

\* ANOVA conducted on the log (x + 1).

### 3.2. 'Rain Shelter and Fungicide Use Reduction' Trial

The rain shelters provided a significant reduction in the duration of the leaf wetness duration in the canopies compared to the unsheltered trees (Table 6). The reduction in wetting duration was equal (2017) or greater (2016 and 2018) than 50% over the months of March and April. The reduction in wetting duration was greater during dew events than it was for rain events. The only exception was for the month of March in 2017, where the reduction was greater during rain events. Rain protection modified very slightly (<5%) the average temperatures and humidity in the canopies. The protection tended to increase the air temperature and reduce the air relative humidity.

**Table 6.** Averaged relative changes in leaf wetness duration, air temperature and air relative humidity in the tree canopy between sheltered trees and unsheltered trees during periods of susceptibility to infection in the 'Rain shelters and fungicide use reduction' trial in 2016, 2017 and 2018.

| Year | Month | Air Relative Humidity | Air Temperature | Changes in Leaf Wetness Duration | | |
|------|-------|----------------------|-----------------|------|-----|-------|
| | | | | Rain | Dew | Total |
| 2016 | March | No data | No data | −45% | −73% | −67% |
| | April | −0.5% | 1.0% | −34% | −81% | −65% |
| 2017 | March | −1.9% | −0.3% | −56% | −35% | −49% |
| | April | 0.5% | 0.3% | −46% | −92% | −50% |
| 2018 | March | −0.2% | 5.1% | −41% | −76% | −56% |
| | April | −0.7% | 3.7% | −49% | −92% | −57% |

In the 'Fungicide reference' mode, three fungicide applications were carried out in 2016 and 2018, compared to only two in 2017. In the 'Rain shelter' mode, the third fungicide application was not used in 2016 and 2018 as planned by the decision rules (Table 2). Over the three years of trials, the fungicide protection was very effective in the 'Fungicide reference' mode, with only a few moniliated twigs per tree (Table 7).

**Table 7.** Number of moniliated twigs per tree (average of the five varieties) and percentage reduction in the number of moniliated twigs in 'Rain shelter' mode compared to 'Fungicide reference' mode in the 'Rain shelter and fungicide use reduction' trial in 2016, 2017 and 2018.

| Date | Number of Moniliated Twigs per Tree | | Probability ($p$) | Reduction Percentage |
|------|------|------|------|------|
| | Fungicide Reference | Rain Shelter | | |
| 28 April 2016 | 0.076 | 0.127 | 0.2228 | No significant differences |
| 7 April 2017 | 0.939 | 0.042 | 0.0023 * | 95.5% |
| 27 April 2018 | 2.3 | 0.8 | 0.0046 * | 65.2% |

* ANOVA conducted on the log (x + 1).

No significant interaction was observed between the 'treatment' and 'variety' factors each year, so the results presented are the averages of the five varieties. In 2016, the orchard's first year of flowering, very few moniliated twigs were observed (approximately one twig in ten trees), and the differences were not significant between the modes 'Fungicide reference' and 'Rain shelter'. In 2017, compared to the mode 'Fungicide reference', a 95% reduction in the number of twigs dried out by *Monilinia* spp. was observed in the mode 'Rain shelter'. This reduction was 65% in 2018 (Table 7).

### 3.3. Organic Farming without Copper Trial

In the organic farming experimental orchard, three fungicide applications took place in the mode 'Organic farming reference' to cover moniliosis contamination in 2021 and 2022 (Table 3).

In 2021, the first year of flowering, very few moniliated twigs were observed in the mode 'Rain shelter', while twig damage was already significant in the 'Control' mode. The reduction in moniliosis damage for the mode 'Rain shelter' compared to the 'Control' was 95% for the Tom Cot® trees and 89% for the Vertige trees (Figure 1). The 'Organic farming reference' with fungicide protection showed more monilia damage than the mode 'Rain shelter' for the Tom Cot® variety (Figure 1). In 2022, twig damage from moniliosis was very high in the 'Control' mode, with 38% of the length of flower-bearing twigs destroyed in the Tom Cot® trees and 68% in the Vertige trees. The 'Organic farming reference' and 'Rain shelter' modes were much less affected with very few damaged twigs in the Tom Cot® trees and about 5% of dried-out flower-bearing twigs in the Vertige trees (Figure 1). The reduction in moniliosis damage for the 'Rain shelter' mode compared to the 'Control' mode was 97% in the Tom Cot® trees and 93% in the Vertige (Figure 1).

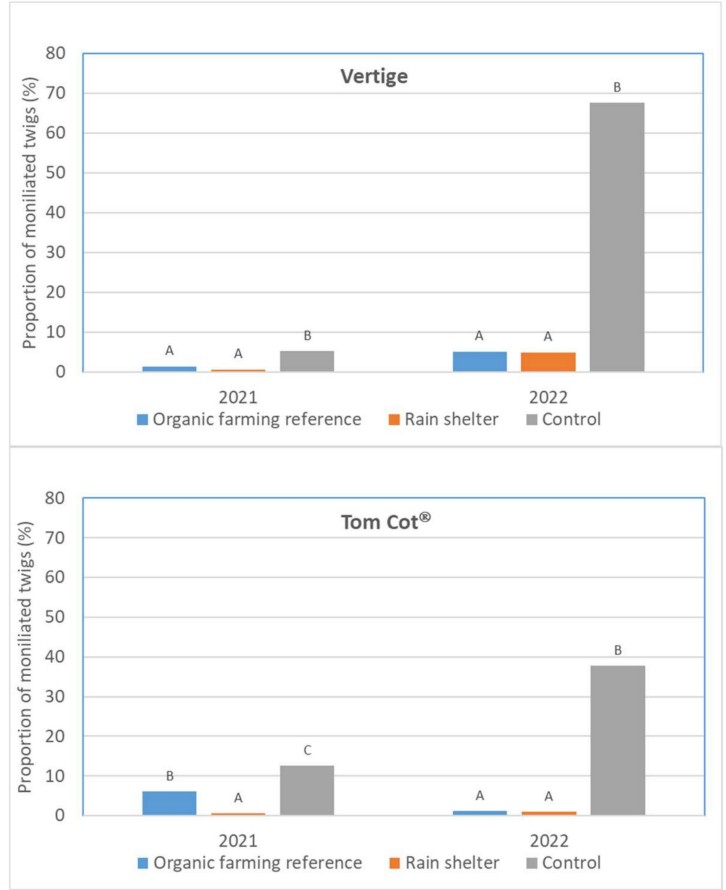

**Figure 1.** Proportion of the length of moniliated flowering twigs in the three modes of the 'Organic orchard without copper' trial in 15 April 2021 and 14 April 2022. For each variety and each year, the linear percentage values of moniliated twigs not presenting the same letters are significantly different at alpha risk = 0.05.

## 4. Discussion

In the first trial, conducted without fungicide protection, the rain protection demonstrated a partial efficacy of 25% to 62% in limiting moniliosis damage on twigs. This relatively low efficacy could be partly due to the fact that the trees were initially pruned in a broad form ('gobelets') before being pruned to form a hedge only 2 m wide. Protection with shelters that were only 3 m wide was perhaps not sufficient. In the two subsequent trials, the trees were pruned in a 'palmette' pattern to form a fruit hedge just 1 m wide, over which protection with 3 m wide rain shelters was installed. In 2017, the two modes tested received exactly the same fungicide protection, and a fairly substantial reduction in the number of moniliated twigs of 95% was observed in the 'Rain shelter' mode. In 2021 and 2022, the strong efficacy of rain covers was also observed, with an 89% to 97% reduction in moniliosis damage. This significant reduction in the number of affected twigs under rain shelters is certainly linked, in part, to the good protection against rain provided by cultivation in a narrow fruit hedge.

In 2018, reduced fungicide protection was used in the 'Rain shelter' mode compared to the 'Fungicide reference' mode, but it still demonstrated a 65% reduction in the number of moniliated twigs. It can be hypothesised that partial-effect cultural methods are more effective when combined with fungicide protection. For example, Didelot et al. [18] showed that varietal mixtures of apple trees combining a variety resistant to scab (due to *V. inaequalis*) with a susceptible variety was more effective when combined with reduced fungicide protection. In their study, the reduction in the incidence of leaf scab in the susceptible variety

grown in a mixture compared to the susceptible variety as a single crop was 7% to 20% in the absence of fungicide protection and 41% to 75% with reduced fungicide protection.

As expected, the rain shelters significantly reduced the leaf wetness duration (LWD) measured within the foliage compared to the unsheltered trees (a reduction of 43% to 67%). The rain shelters reduced wetting times both during and after a rain event and dew deposition times after a clear night. Overall, the rain shelters led to a greater reduction in dewy conditions than in rainy ones (Tables 4 and 6). This is consistent with the LWD observed in the tree canopies [19] and the physics of water deposition by rain and dew [20]. Indeed, a rain shelter acts as an effective physical barrier against water droplets from rain but also prevents leaves from night-time cooling by limiting longwave energy losses during clear sky conditions. However, rain shelters also limit the evaporation process by decreasing net radiation [12] and then increasing evaporation time [20]. The sensors were located at different locations within the canopies and it has been shown that the tree canopies intercept part of the rain droplets, and then lead to a reduction in the LWD within the canopies [21]. This means part of the observed reduction in the LWD may also result from a canopy effect.

Very few differences were observed in the daily averaged air temperature (up to 6%) and the daily averaged air relative humidity (up to 1%). So, the overall decrease in 'damage' in trials one and two may be mainly due to the reduced LWD as a result of the use of rain shelters and not by changes in the air properties. The rain protection, by limiting the direct impact of raindrops on the tree canopies, could also limit the dispersion of conidia present on cankers and mummified fruits.

In the third experimental approach, managed according to organic principles, the levels of moniliosis damage on the twigs of the system with rain protection and no fungicide protection are lower or equivalent to the organic reference system using fungicide protection. These results make it possible to envisage organic apricot production without the use of cupric fungicides [22].

Rain protection has also been shown to be highly effective in reducing the development of rust epidemics in apricot trees by limiting primary contamination in May and June [23,24]. However, it is difficult to assess the direct effectiveness of rain shelters on moniliosis on fruits at harvest. Indeed, because apricot fruits are not very sensitive to moniliosis, the rotten fruits found at harvest are often those that have suffered damage (earwig bites, hail impact, etc.). However, the installation of rain shelters requires numerous posts and anchor cables which allows earwigs to pass towards the canopy of trees and so limits the effectiveness of physical barriers such as the installation of glue bands on the trunk. On the other hand, rain shelters greatly reduce fruit damage due to hail.

Providing rain protection could be an interesting solution for producing organic apricots without resorting to copper-based fungicides, the environmental impact of which is not insignificant [25,26].

## 5. Conclusions

To our knowledge, this is the first study to assess the effect of rain shelters on the development of twig blight on stone fruits. The efficiencies observed showed that these rain shelters could be an effective solution to protect trees from moniliosis damage, thus reducing the use of fungicides. Our results support that the wetness duration is the main microclimatic factor that is reduced by rain shelters. Our observations indicate that, under the conditions of our trials, the rain covers used did not significantly affect fruit production. However, the evaluation of these rain shelters on tree growth, productivity and fruit quality should be continued. The installation of rain shelters in apricot orchards can be expensive, and the profitability of such orchards will have to be evaluated in different situations according to input and labour costs and the selling price of apricots produced.

**Author Contributions:** Conceptualisation, L.B., F.C., C.G., P.W. and M.S.; methodology, L.B., F.C., C.G., P.W. and M.S.; validation, L.B., F.C., C.G., P.W. and M.S; resources, L.B., F.C., C.G., P.W. and M.S.; data curation, L.B., P.W. and M.S.; writing—original draft preparation, L.B.; writing—review and editing, L.B. and M.S.; supervising, L.B. and M.S.; funding acquisition, L.B. and M.S. All authors have read and agreed to the published version of the manuscript.

**Funding:** This research was funded by the French Ecophyto Programmes (Ministry of Agriculture: 2013—X2CT75AR; Ministry of Agriculture: 2019—X4IN40AR).

**Data Availability Statement:** Not applicable.

**Acknowledgments:** The authors thank Franck Merlin for the fungicide applications, and Pédro Asencio, Marie Gaslain and Frédéric Oboussier for their technical help.

**Conflicts of Interest:** The authors declare no conflict of interest.

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
