# Peer review of "Protecting Apricot Orchards with Rain Shelters Reduces Twig Blight Damage Caused by Monilinia spp. and Makes It Possible to Reduce Fungicide Use"

_agronomy, doi:10.3390/agronomy13051338_

Round 1

Reviewer 1 Report

The authors evaluated the influence of transparent rain shelters on the modification of the microclimate at the level of the tree canopy and on the reduction of moniliosis damage to twigs, and also tested the effect of transparent rain shelters on fungicide use reduction. This research is unique and innovative to some extent.

1.     The paper is difficult to read and the graphs presented are not easy to understand. Please revise the full text to make it easy to understand.

2.     There are some puzzling aspects of the design, such as whether the identification of pathogens is meaningful in the text?The objective of this paper is to study the effect of rain shelter culture on brown rot. Does the change of microenvironment affect the pathogen species? Is this the author's purpose? But for the purposes of this study, such work seems unnecessary.

3.     The experimental design lacks rigor and needs to be improved. The equipment models and monitoring heights used to monitor the meteorological data of the rain shelter and outdoor open-air are different. In particular, the meteorological data collected are not analyzed in depth, but are compared directly on a monthly basis. The results may not be able to find the internal relationship.

4.     How to calculate the leaf wetting time mentioned in the article? What is the reading of leaf moisture that can be considered as leaf moisture? Please explain.

Author Response

The manuscript was translated into English by Andrew Lever (Argoat communications), a professional translator whose quality of scientific translations is recognized by INRAE.

1. Changes have been made in the structuring of the text and the presentation of the results.

2. We deleted paragraph 3.1 on the identification of the species of Monilinia present in the experimental orchards. And we inserted a simple sentence on this identification in the following paragraph.

We disagree with your remark and we think that these microclimatic measurements were necessary for two main reasons at least:

a) Firstly, microclimatic conditions, and in particular air temperature, air relative humidity and the duration of wetting of organs, strongly condition the development of fungal diseases.

b) Secondly, rain shelters aim as intercepting raindrops. Therefore it was necessary to provide an estimation of the interception efficiency of the shelter by measuring some microclimatic conditions. Thanks to these measurements, we show that the reduction in contamination by Monilinia spp. is mainly due to the reduction in wetness duration.

3. As written in § 2.4 Climatic conditions and canopy microclimate, all sensors were placed in tree canopies at a height of 1.8 meters

The objective of this research work was not to analyze data in depth and internal relationships between air properties and wetness duration but rather to estimate the effect of rain shelter on water deposit on canopy including dew and rain. Therefore, we think that a monthly basis is sufficient to show the effect of shelter and is consistent with disease notation.

4. The estimation of the LWD was done as follow. First and prior to the experiment, the resistivity (in ohms) of the sensor was estimated in controlled conditions for dry and wet conditions. A threshold value was then obtained that enables to indicate whether liquid water is present (LWD is set to 1) or not (LWD is set to zero) at the sensor surface. In practice and in the experiment, each 10 minutes, the LWD was estimated by converting the electrical resistance measured by the sensors in 0 and 1 according the threshold value. As this value standed for a 10 minutes measured, this value was multiplied by 10 minutes to get the LWD. The procedure was applied to the entire dataset to obtained LWD every 10 minutes. The values were summed to obtain the monthly values. We added this explanation in line 212.

We agree with you that the LWD duration estimated from sensors does not equal to the real leaf moisture which may depend on many their age, orientation and surface rugosity. However, this is not a problem is this work. Indeed the objective was not to measure the real leaf moisture but to estimate the effect of shelter on the water availability at the crown level. The use of a sensor allows us to get a common and standardized values that avoid the variability in real leaf moisture in a tree crown.

Reviewer 2 Report

No suggestions or comments to the authors is a good article ready for  publication 

Author Response

The manuscript was translated into English by Andrew Lever (Argoat communications), a professional translator whose quality of scientific translations is recognized by INRAE.

Reviewer 3 Report

The widespread of diseases that caused by Monilinia spp with regard to the possible negative effects on fruit production of pome and stone fruit,  especially in temperate zone. Herein lies the value of this study. The paper evaluates the effect of rain shelters on twig blight that caused by Monilinia spp., for apricots tree. The results showed the effective method that could reduce twig blight and following fungicidal use. The MS was written in a good scientific way. The experimental design was also implemented  in scientific method. The results is clearly presented with good interpretation

But, the most important question is, the experiment was carried on apricot trees that are 25 years old, why the author didn't mention the effect of treatments on vegetative growth, trees yield, productivity and fruit quality?  

summary and conclusion is missed.

Author Response

The manuscript was translated into English by Andrew Lever (Argoat communications), a professional translator whose quality of scientific translations is recognized by INRAE.

In the first trial, the apricot trees were 17 to 18 years old in 2014 and 2015. In the second trial, the apricot trees were 2 to 4 years old from 2016 to 2018. In the third trial, the apricot trees were 2 to 3 years old in 2021 and 2022. The planting dates of the different orchards are indicated in Materials and Methods.

We have, in fact, recorded data on tree growth, productivity and the development of other pests. But the presentation of these results did not seem suitable for this special issue on Monilinia, and would have led to the production of an article that was too voluminous.

Reviewer 4 Report

Review of the article Agronomy-2331659. March 2023.

Outstanding work that produces information on the control of Monilinia in different varieties of apricot tree, years and environmental conditions, through the use of rain shelters.

I am thinking that to improve the article, the authors should clarify some details of the text:- Document a little more the type of mesh used.

- In each experiment, they said when the trees are covered (lines 127, 141-142, 169), but it would be convenient to clarify (to say) if they remove the shelters and when they do it. At least in the text it is not clear.

- If the study is focused on the effect of the rain shelters, the authors should justify why they have incorporated the section 3.1, either expanding the discussion of the results (since they present them) or better (in my opinion) directly eliminating this small section 3.1. I understand that its removal does not affect the rest of the study.

Author Response

- We have indicated the composition of the rain cover in section 2.1.

- We specified in section 2.1 the removing dates of the rain covers. « They were folded up in July after fruit harvesting ».

- «…or better (in my opinion) directly eliminating this small section 3.1. » We deleted paragraph 3.1 on the identification of the species of Monilinia present in the experimental orchards. And we inserted a simple sentence on this identification in the following paragraph.

-"Are the methods adequately described?" We have added explanations concerning the LWD estimation method (line 212 to 220).

Round 2

Reviewer 3 Report

Dear author,

The conclusion is still missed

regards, 

Author Response

We have added a concluding paragraph. The last sentence of the discussion of the previous version of the manuscript has been moved to the end of this concluding paragraph.